# An Order Reduction Design Framework for Higher-Order Binary Markov Random Fields

**DOI:** 10.3390/e25030535

**Published:** 2023-03-20

**Authors:** Zhuo Chen, Hongyu Yang, Yanli Liu

**Affiliations:** 1Sichuan University National Key Laboratory of Fundamental Science on Synthetic Vision, Sichuan University, No. 29 Wangjiang Road, Chengdu 610065, China; zhuochen1989@outlook.com; 2College of Computer Science, Sichuan University, No. 29 Wangjiang Road, Chengdu 610065, China

**Keywords:** Markov random field, discrete optimization, energy minimization, order reduction method, higher-order binary MRF, image desnoising

## Abstract

The order reduction method is an important approach to optimize *higher-order binary Markov random fields* (HoMRFs), which are widely used in information theory, machine learning and image analysis. It transforms an HoMRF into an equivalent and easier *reduced first-order binary Markov* random field (RMRF) by elaborately setting the coefficients and auxiliary variables of RMRF. However, designing order reduction methods is difficult, and no previous study has investigated this design issue. In this paper, we propose an order reduction design framework to study this problem for the first time. Through study, we find that the design difficulty mainly lies in that the coefficients and variables of RMRF must be set simultaneously. Therefore, the proposed framework decomposes the design difficulty into two processes, and each process mainly considers the coefficients or auxiliary variables of RMRF. Some valuable properties are also proven. Based on our framework, a new family of 14 order reduction methods is provided. Experiments, such as synthetic data and image denoising, demonstrate the superiority of our method.

## 1. Introduction

The *higher-order binary Markov random field* (HoMRF) is a non-convex optimization model widely used in the fields of economy, information theory, quantum computing, machine learning and image analysis [1,2,3,4,5,6,7,8,9,10,11,12,13,14]. Recently, a new order reduction method has been developed to optimize HoMRF energies. The order reduction method transforms the HoMRF into a *reduced quadratic binary Markov random field* (RMRF) by elaborately setting the coefficients and auxiliary variables of RMRF to ensure the equivalence between HoMRF and RMRF. By equivalently transforming a complex HoMRF into an easier RMRF, the order reduction method achieves remarkable performance and thus has attracted increasing attention from researchers.

Since it is difficult to reduce an HoMRF into an RMRF in a single step, the mainstream approach is to transform a higher-order monomial into a sum of linear and quadratic monomials, and then iteratively perform this operation for all higher-order monomials of the HoMRF until a RMRF is obtained. According to the higher-order monomials to be reduced, previous order reduction methods can be divided into two categories: the methods of reducing higher-order negative monomials, and the methods of reducing higher-order positive monomials. In first category, Freedman et al. [15] developed an order reduction method for higher-order negative monomials for the first time. Then, Anthony et al. [16,17,18] proposed an order reduction method that is a more general form of the method [15]. Both two methods require only a binary auxiliary variable to produce submodular monomials, which can be easily minimized in polynomial time [19]. Anthony et al. [16,17,18] and Yip et al. [20] also developed two alternative methods to reduce higher-order negative monomials. Since these two methods produce nonsubmodular monomials that make minimizing RMRF to be an NP-hard problem, they are rarely used in practice.

Compared with reducing higher-order negative monomials, designing the order reduction method for higher-order positive monomials is much more difficult [1,2,16,17,18,21,22]. Ishikawa [1,2] firstly developed a method of reducing higher-order positive monomials, by summarizing the rules in a large number of order reduction methods from the third-order to seventh-order positive monomials. Further, Anthony et al. [16,17,18] initiated a systematic study of lower and upper bounds on the number of binary auxiliary variables required by the order reduction methods. Nevertheless, Anthony et al. [16,17,18] only proved the existence of these upper and lower bounds, and failed to design any specific order reduction method that matches the corresponding upper and lower bounds. Until recently, Boros et al. [21,22] provided four order reduction methods, which utilize less auxiliary variables than Ishikawa [1,2]. Compared with the method of reducing higher-order negative monomial, the method of reducing higher-order positive monomials suffer from two critical problems: the nonsubmodular quadratic monomials obtained lead to the NP-hard problem of minimizing RMRF energies, and the introduced multiple binary auxiliary variables increase the computational complexity.

Therefore, designing more powerful order reduction methods for higher-order positive monomials is important and challenging. We observe that the main obstacle of designing order reduction methods for higher-order positive monomials is the interdependence between the coefficients and auxiliary variables of RMRF. Studies [21,22] also noticed the interdependence of auxiliary variables and coefficients. When designing order reduction methods that use logarithmic auxiliary variables, they proved that the coefficients increase exponentially. However, they did not further research the interdependence issue. Such interdependence forces us to consider the auxiliary variables and coefficients of RMRF simultaneously rather than separately, which greatly increases the design difficulty.

In this paper, we propose a unified design framework to reduce higher-order positive monomials. The proposed framework not only significantly decreases the difficulty of designing order reduction methods, but also provides 14 order reduction methods that allow applications in different fields to pick up their best method. Since the main difficulty of reducing HoMRF is due to the interdependence between the coefficients and auxiliary variables of RMRF, our core idea is to decompose this difficulty into two easier processes, where each process considers the coefficients or auxiliary variables separately. The flowchart of the proposed framework is shown in Figure 1. First, we generalize previous order reduction methods and propose a novel *general reduction function* (GRF) that requires only one auxiliary variable for any higher-order positive monomial. Different from the previous order reduction methods that utilize multiple binary auxiliary variables, our integral auxiliary variable is fixed, hence we can focus on setting the coefficients of RMRF. We rigorously prove the properties of the minimum value range of the integral auxiliary variable. Second, since the coefficients are already considered in the first process, we propose substitution and minimum transformations that convert the integral auxiliary variable of GRF into binary auxiliary variables. With the proposed transformations, we can concentrate on setting the binary auxiliary variables of RMRF. Based on the framework, a new family of 14 order reduction methods is proposed. Moreover, some state-of-the-art reduction methods can be easily derived from the proposed framework. We believe that our novel framework may take the design of order reduction methods to a new level. Comparison experiments show the superiority of our work.

The main contributions of our study are as follows:We propose an order reduction design framework that divides the complex design of order reduction methods into two simpler processes. Compared with existing works, our framework significantly decreases the design difficulty for order reduction methods.A novel GRF is developed to generalize previous order reduction methods. Unlike the previous methods with multiple binary auxiliary variables, GRF utilizes an integral auxiliary variable. Some valuable properties of GRF are also rigorously proved.Two sets of substitution and minimum transformations are developed to produce more order reduction methods. A variety of 14 order reduction methods are produced to enable applications in different fields to choose their most suitable method. Moreover, four state-of-the-art order reduction methods can be easily derived from our work.

## 2. Notation and Related Work

In this section, we provide some basic terminologies, as well as the notations and backgrounds necessary for understanding the subsequent sections.

### 2.1. Higher-Order Binary Markov Random Field

In this paper, we focus on the binary Markov random field, since a Markov random field with any multi-labels can be converted to a binary Markov random field via standard techniques [2,23,24,25,26]. An HoMRF, also known as a pseudo-Boolean function [19,27], is a binary Markov random field whose order is greater than two. It is defined as $E:{\left\{0,1\right\}}^{n}\to R$. Here, *n* is the number of variables and R is the set of real numbers. The energy function *E* can be uniquely expressed as multilinear polynomials [19]:(1)$$E\left(x_{1},\ldots ,x_{n}\right)=\sum_{S\subseteq [n]}c_{S}\prod_{i\in S}x_{i}$$
where $x_{1},\ldots ,x_{n}$ are binary variables, [n]={1,2,…,n}, $c_{S}\in R$ is the coefficient of $c_{S}\prod_{i\in S}x_{i}$. In particular, when S=Ø, $c_{S}$ denotes the constant of $E(x_{1},\ldots ,x_{n})$. The size of *S* is denoted by |S|. If |S|>2, the monomial $c_{S}\prod_{i\in S}x_{i}$ is a higher-order monomial. For brevity, a general higher-order negative monomial is denoted as $-x_{1}x_{2}\cdots x_{\left|S\right|}$, and a general higher-order positive monomial is denoted as $x_{1}x_{2}\cdots x_{\left|S\right|}$, 2<|S|⩽n. Denote the complement of $x_{i}$ as ${\bar{x}}_{i}=1-x_{i},i\in \left[n\right]$.

Although minimizing a general energy function is generally an NP-hard problem [19,28], the submodular energy function can be minimized globally in polynomial time. Proposition 24 in [19] is commonly used to judge whether *E* satisfies submodular.

As the higher orders make an HoMRF more difficult to optimize, the study [29] first proposes a substitution-based method to transform an HoMRF into a *reduced binary Markov random field* (RMRF). However, there are already experiments showing its poor performance [1,2]. Although some works can effectively reduce an HoMRF [27,30,31,32,33,34,35] or minimize an HoMRF [36,37,38,39,40,41,42,43], they can only be applied to some special HoMRFs, which limits their applications. In this paper, we focus on reducing the general HoMRF. In reference [18], the RMRF of an HoMRF is formally defined as follows.

**Definition 1.** 
*Given an HoMRF $E(x_{1},\ldots ,x_{n})$, R is an RMRF of $E(x_{1},\ldots ,x_{n})$ if $R:{\left\{0,1\right\}}^{n}\times {\left\{0,1\right\}}^{l}\to R$ is a quadratic polynomial depending on $x_{1},\ldots ,x_{n}$ and on l auxiliary variables $y_{1},\ldots ,y_{l}$ such that*

(2)
$$E\left(x_{1},\ldots ,x_{n}\right)=\min_{y_{1},\ldots ,y_{l}\in {\left\{0,1\right\}}^{m}}R\left(x_{1},\ldots ,x_{n},y_{1},\ldots ,y_{l}\right),\;\forall x_{1},\ldots ,x_{n}\in {\left\{0,1\right\}}^{n}.$$



### 2.2. Related Works of Order Reduction Methods

As shown in Definition 1, the goal of reducing the HoMRF *E* is to design an equivalent RMRF *R* for *E*. However, directly reducing *E* into *R* is extremely difficult. A mainstream order reduction method is to transform the higher-order monomials of *E* one by one until *R* is obtained. Note that an important property of the higher-order monomial $c_{S}x_{1}x_{2}\cdots x_{\left|S\right|}$ is symmetry, which means that it is invariant under any permutation of the coordinates {1,2,…,m} of the variable *x*. Most of reduction methods [16,18,21,22] utilize the symmetry to reduce higher-order monomials based on the following concept:

**Definition 2.** 
*A function $f\left(x,y\right):{\left\{0,1\right\}}^{\left|S\right|}\times {\left\{0,1\right\}}^{m}\to R$ is called the reduction function (RF) of $-x_{1}x_{2}\cdots x_{\left|S\right|}$ or $x_{1}x_{2}\cdots x_{\left|S\right|}$ if it is a symmetric quadratic polynomial about $x_{1},x_{2},\ldots ,x_{\left|S\right|}$ and satisfies*

(3)
$$-x_{1}x_{2}\cdots x_{\left|S\right|}=\min_{y\in {\left\{0,1\right\}}^{m}}f\left(x,y\right)$$

*or*

(4)
$$x_{1}x_{2}\cdots x_{\left|S\right|}=\min_{y\in {\left\{0,1\right\}}^{m}}f\left(x,y\right)$$

*where $x={\left(x_{1},x_{2},\ldots ,x_{\left|S\right|}\right)}^{⊤}$ and $y={\left(y_{1},y_{2},\ldots ,y_{m}\right)}^{⊤}$.*


We introduce the order reduction methods of higher-order negative terms and higher-order positive terms, respectively. First, for the higher-order negative monomial, Freedman et al. [15] are the first to provide:(5)$$-x_{1}x_{2}\cdots x_{\left|S\right|}=\min_{y\in \{0,1\}}\left(|S\right|-1-\sum_{i=1}^{\left|S\right|}x_{i})y.$$This method produces a submodular [19] RF and requires only one auxiliary variable. Thus, it achieves excellent experimental performance [1,2,3,4]. A more general form of Equation (Equation 5) is proposed by paper [17]:(6)$$-x_{1}x_{2}\cdots x_{\left|S\right|}=\min_{y\in \{0,1\}}\left(c|S\right|-1-c\sum_{i=1}^{\left|S\right|}x_{i})y$$
where c⩾1 is the coefficient. To reduce the higher-order negative monomial, Anthony et al. [18] provide another method with one auxiliary variable, but their method do not satisfy submodular; alternatively, Yip et al. [20] propose a method that is an equivalent transformation of Equation (Equation 5).

Second, to reduce the higher-order positive monomial, order reduction method generates a nonsubmodular RF and requires multiple auxiliary variables, which is the main reason for the difficulty in minimizing the RMRF. A lot of efforts have been devoted to designing the RF for a higher-order positive monomial. Ishikawa [1,2] observes the RF of $x_{1}x_{2}x_{3},x_{1}x_{2}x_{3}x_{4},\ldots ,x_{1}x_{2}x_{3}x_{4}x_{5}x_{6}x_{7}$ and then summarizes the first order reduction method *high order clique reduction* (HOCR) for a general higher-order positive monomial:(7)$$x_{1}x_{2}\cdots x_{\left|S\right|}=\min_{y_{1},\ldots ,y_{m}\in {\left\{0,1\right\}}^{m}}\sum_{i=1}^{m}(c_{i,|S|}(2i-\sum_{i=1}^{\left|S\right|}x_{i})-1)y_{i}+\sum_{1⩽i<j⩽|S|}x_{i}x_{j}$$
where
$$m=\left\lfloor \frac{|S|-1}{2}\right\rfloor ,c_{i,|S|}=\left\{\begin{array}{cc} 1, & \left|S\right|\;\mathrm{is}\;\mathrm{odd}\;\mathrm{and}\;m=i,\\ 2, & \mathrm{otherwise}. \end{array}\right.$$The researches [21,22] systematically study the upper and lower bounds of the number of auxiliary variables for several classes of specially structured pseudo-Boolean functions, and provide two order reduction methods that require a logarithmic number of auxiliary variables. One is the *logarithmic reduction type-1* (LogR-1) method:(8)$$x_{1}x_{2}\cdots x_{\left|S\right|}=\min_{y_{0},\ldots ,y_{m-1}\in \left\{0,1\right\}}(c+\sum_{i=1}^{\left|S\right|}x_{i}-\sum_{i=0}^{m-1}2^{i}y_{i}{)}^{2}$$
where $m=\lceil \log_{2}\left(|S|)\right\rceil$ and $c=2^{m}-\left|S\right|$. Another is the *logarithmic reduction type-2* (LogR-2) method
(9)$$x_{1}x_{2}\cdots x_{\left|S\right|}=\min_{y_{1},\ldots ,y_{m}\in \left\{0,1\right\}}\frac{1}{2}(c+\sum_{i=1}^{\left|S\right|}x_{i}-\sum_{i=1}^{m}2^{i}y_{i})(c+\sum_{i=1}^{\left|S\right|}x_{i}-\sum_{i=1}^{m}2^{i}y_{i}-1)$$
where $m=\lceil \log_{2}\left(|S|)\right\rceil -1$ and $c=2^{m+1}-\left|S\right|$. However, they notice that Equations (Equation 8) and (Equation 9) require exponential coefficients, which have a negative impact on computational performance. Thus, the following *linear reduction* (LinR) method is developed:(10)$$x_{1}x_{2}\cdots x_{\left|S\right|}=\min_{y_{1},\ldots ,y_{m}\in \left\{0,1\right\}}\frac{1}{2}(\sum_{i=1}^{\left|S\right|}x_{i}-cy_{1}-2\sum_{i=2}^{m}y_{i})(\sum_{i=1}^{\left|S\right|}x_{i}-cy_{1}-2\sum_{i=2}^{m}y_{i}-1)$$
where $\frac{\left|S\right|}{4}⩽m⩽\frac{\left|S\right|}{2}$ and c=|S|−2m. Finally, they propose a *square root reduction* (SqrtR) method that nicely matches the lower bound in [17]:(11)$$\begin{array}{cc} x_{1}x_{2}\cdots x_{\left|S\right|}=\min_{\begin{array}{c} y_{0},\ldots ,y_{c-1}\in \left\{0,1\right\}\\ y_{0}^{\prime },\ldots ,y_{c-1}^{\prime }\in \left\{0,1\right\} \end{array}}(y_{i_{0}}y_{i_{0}}^{\prime } & +\lambda ({\left(1-\sum_{i=0}^{c-1}y_{i}\right)}^{2}+{\left(1-\sum_{j=0}^{c-1}y_{j}^{\prime }\right)}^{2})\\ \; & +\lambda {\left(\sum_{i=1}^{\left|S\right|}x_{i}-\left(c\sum_{i=0}^{c-1}i\cdot y_{i}+\sum_{j=0}^{c-1}j\cdot y_{j}^{\prime }\right)\right)}^{2}) \end{array}$$
where $m=2\lceil \sqrt{|S|+1}\;\rceil$ is the number of auxiliary variables, $c=\frac{m}{2}$, and λ is a large number. $y_{i_{0}}$ and $y_{i_{0}}^{\prime }$ must satisfy the condition $c\cdot i_{0}\cdot y_{i_{0}}+j_{0}\cdot y_{j_{0}}^{\prime }=\left|S\right|$.

## 3. The First Process of the Proposed Framework: General Reduction Function (GRF)

In this section, we define a novel reduction function g(x,z) and demonstrate the convenience of designing it in Section 3.1. Then, we prove some critical properties of g(x,z) in Section 3.2.

### 3.1. General Reduction Function (GRF)

Based on Definition 2, setting the coefficients and auxiliary variables of RMRF is greatly simplified to designing f(x,y). However, designing f(x,y) is still very difficult, mostly owing to the interdependence between the coefficients and the auxiliary variables of f(x,y). Such interdependence has rarely been studied, yet it must be considered when designing f(x,y). Therefore, to avoid considering both coefficients and auxiliary variables, we propose a novel function g(x,z) with a fixed auxiliary variable *z*, which allows us to focus on designing the coefficient.

**Definition 3.** 
*A function $g\left(x,z\right):{\left\{0,1\right\}}^{\left|S\right|}\times W\to R$ is called the general reduction function (GRF) of $x_{1}x_{2}\cdots x_{\left|S\right|}$ if it is a symmetric polynomial about $x_{1},x_{2},\ldots ,x_{\left|S\right|}$ and satisfies*

(12)
$$x_{1}x_{2}\cdots x_{\left|S\right|}=\min_{z\in W}g\left(x,z\right)$$

*where $x={\left(x_{1},x_{2},\ldots ,x_{\left|S\right|}\right)}^{⊤}$; W={0,1,…,|W|−1} is the value space of z and |W| is the size of W.*


Since the higher-order negative monomial has the excellent reduction method Equations (Equation 5) and (Equation 6), Definition 3 only considers the higher-order positive monomial. Furthermore, g(x,z) relaxes the restriction of f(x,y) in two ways: g(x,z) can be more than quadratic and *z* has not be binary. Since g(x,z) is symmetric about $x_{1},\ldots ,x_{\left|S\right|}$, we denote an auxiliary function h:{0,1,…,|S|}×W→R of g(x,z):(13)$$\begin{array}{cc} g(x,z)= & g_{1}\left(z\right)+\sum_{i=1}^{\left|S\right|}g_{2}\left(z\right)x_{i}+\sum_{1⩽i<j⩽\left|S\right|}cx_{i}x_{j}\\ = & g_{1}\left(z\right)+g_{2}\left(z\right)k+c\theta \left(k\right)=h\left(k,z\right) \end{array}$$
where $g_{1}:W\to R$ and $g_{2}:W\to R$ are polynomials; $k=\sum_{i=1}^{\left|S\right|}x_{i}$ and θ(k)=k(k−1)/2; c∈R is the coefficient.

Since g(x,z) is a GRF of $x_{1}x_{2}\cdots x_{\left|S\right|}$, then $x_{1}x_{2}\cdots x_{\left|S\right|}=\min_{z\in W}h\left(k,z\right)=𝟙\left(k=|S|\right)$, where the indicator function 𝟙(k=|S|)=1⇔k=|S|. Obviously, h(k,z) is simple with only two variables. Thus, we can easily utilize it to formulate three specific forms of g(x,z).

Let $W_{1}=\left\{0,1,\ldots ,|S|\right\}$. Then, the *general reduction type-1* (GR-1) method is
(14)$$x_{1}x_{2}\cdots x_{\left|S\right|}=\min_{z\in W_{1}}(𝟙(z=|S|)+\lambda ∥z-x_{1}-\cdots -x_{\left|S\right|}{∥}_{2}^{2})$$
where ${∥\cdot ∥}_{2}$ is the 2-norm, λ>1 is a large number and the indicator function 𝟙(z=|S|)=1⇔z=|S|.Let $W_{2}=\left\{0,1,\ldots ,|S|-1\right\}$. Then, the *general reduction type-2* (GR-2) method is
(15)$$x_{1}x_{2}\cdots x_{\left|S\right|}=\min_{z\in W_{2}}{∥z-x_{1}-\cdots -x_{\left|S\right|}∥}_{2}^{2}.$$Let $W_{3}=\left\{0,1,\ldots ,\left\lfloor \frac{|S|-1}{2}\right\rfloor \right\}$. Then, the *general reduction type-3* (GR-3) method is
(16)$$x_{1}x_{2}\cdots x_{\left|S\right|}=\min_{z\in W_{3}}\frac{1}{2}(2z-\sum_{i=1}^{\left|S\right|}x_{i})(2z+{\left(-1\right)}^{\left|S\right|}-\sum_{i=1}^{\left|S\right|}x_{i}).$$

### 3.2. Properties of GRF

The computational complexity of minimization algorithms, such as the *sequential tree-reweighted algorithm* (TRW-S) [44,45], depend heavily on the size of W. Therefore, it is natural to hope that the size of W is as small as possible. Before the discussion, we prove some important properties of g(x,z).

**Theorem 1.** 
*Let g(x,z) be the GRF of $x_{1}x_{2}\cdots x_{\left|S\right|}$ and h(k,z) be the auxiliary function of g(x,z).*
*1.* 
*$g_{1}\left(z\right)⩾0$ and $g_{1}\left(z\right)≢0$;*
*2.* 
*c>0;*
*3.* 
*$g_{2}\left(z\right)⩽0$.*



**Proof.** 
1.Prove that $g_{1}\left(z\right)⩾0$. Based on the definition of h(k,z), $\min_{z\in W}h\left(k,z\right)=𝟙\left(k=|S|\right)$. If k=0, then $\min_{z\in W}h\left(0,z\right)=0$. According to Equation (Equation 13), $h\left(0,z\right)=g_{1}\left(z\right)$. Thus, $\min_{z\in W}h\left(0,z\right)=\min_{z\in W}g_{1}\left(z\right)=0\Rightarrow g_{1}\left(z\right)⩾0$.2.Prove that c>0. Since $\min_{z\in W}h\left(k,z\right)=𝟙\left(k=|S|\right)$, if k=1, then $\min_{z\in W}h\left(1,z\right)=0$. According to Equation (Equation 13), $h\left(1,z\right)=g_{1}\left(z\right)+g_{2}\left(z\right)$. Suppose an integral $z_{0}\in W$ such that
$$\min_{z\in W}h\left(1,z\right)=\min_{z\in W}(g_{1}\left(z\right)+g_{2}\left(z\right))=g_{2}\left(z_{0}\right)+g_{1}\left(z_{0}\right)=0\Rightarrow g_{2}\left(z_{0}\right)=-g_{1}\left(z_{0}\right).$$Similarly, if k=|S|, then $\min_{z\in W}h\left(|S|,z\right)=1$. When $z=z_{0}$, we have
$$h(|S|,z_{0})⩾\min_{z\in W}h\left(|S|,z\right)=1.$$According to Equation (Equation 13), the above inequality is
$$g_{1}\left(z_{0}\right)+\left|S\right|g_{2}\left(z_{0}\right)+\frac{\left|S|(|S|-1\right)}{2}c⩾1.$$Since $g_{2}\left(z_{0}\right)=-g_{1}\left(z_{0}\right)$, $g_{1}\left(z_{0}\right)⩾0$ and |S|>2, the above inequality is transformed as
$$\begin{array}{cc} \; & g_{1}\left(z_{0}\right)-\left|S\right|g_{1}\left(z_{0}\right)+\frac{\left|S|(|S|-1\right)}{2}c⩾1\\ \Rightarrow & \frac{\left|S|(|S|-1\right)}{2}c-\left(|S|-1\right)g_{1}\left(z_{0}\right)⩾1\\ \Rightarrow & c⩾\frac{2+2\left(|S|-1\right)g_{1}\left(z_{0}\right)}{\left|S|(|S|-1\right)}>0. \end{array}$$3.Prove that $g_{2}\left(z\right)⩽0$. We prove this case by contradiction. Suppose $z=z_{0}\in W$ such that $g_{2}\left(z_{0}\right)>0$ and $\min_{z\in W}h\left(k,z\right)=h\left(k,z_{0}\right)=𝟙\left(k=|S|\right)$. We discuss it in two cases according to the value of *k*.(a)If 1⩽k<|S|, then $h(k,z_{0})=0$. According to Equation (Equation 13), we have
$$\begin{array}{cc} \; & h\left(k,z_{0}\right)=g_{1}\left(z_{0}\right)+kg_{2}\left(z_{0}\right)+\frac{k(k-1)}{2}c=0\\ \Rightarrow & g_{2}\left(z_{0}\right)=-\frac{1}{k}g_{1}\left(z_{0}\right)-\frac{k-1}{2}c. \end{array}$$
where 1⩽k<|S|. Since $g_{1}\left(z_{0}\right)⩾0$ and c>0, then $g_{2}\left(z_{0}\right)<0$, contradicting the assumption that $g_{2}\left(z_{0}\right)>0$.(b)If k=|S|, then $h(|S|,z_{0})=1$. Suppose $z_{0}\neq z_{1}\in W$ such that
$$\min_{z\in W}h\left(1,z\right)=h\left(1,z_{1}\right)=g_{2}\left(z_{1}\right)+g_{1}\left(z_{1}\right)=0\Rightarrow g_{2}\left(z_{1}\right)=-g_{1}\left(z_{1}\right).$$Directly from the definition of h(k,z), we can write
$$\begin{array}{cccc} \; & \; & h(|S|,z_{1}) & ⩾\min_{z\in W}h\left(|S|,z\right)=h(|S|,z_{0})\\ \; & \Rightarrow & g_{1}\left(z_{1}\right)+\left|S\right|g_{2}\left(z_{1}\right)+\frac{\left|S|(|S|-1\right)}{2}c & ⩾g_{1}\left(z_{0}\right)+\left|S\right|g_{2}\left(z_{0}\right)+\frac{\left|S|(|S|-1\right)}{2}c\\ \; & \Rightarrow & g_{1}\left(z_{1}\right)+\left|S\right|g_{2}\left(z_{1}\right) & ⩾g_{1}\left(z_{0}\right)+\left|S\right|g_{2}\left(z_{0}\right). \end{array}$$Since |S|>2, $g_{2}\left(z_{1}\right)=-g_{1}\left(z_{1}\right)$, $g_{1}\left(z\right)⩾0$ and $g_{2}\left(z_{0}\right)>0$, we have
$$\begin{array}{cc} \; & \left(1-|S|\right)g_{1}\left(z_{1}\right)⩾g_{1}\left(z_{0}\right)+\left|S\right|g_{2}\left(z_{0}\right)\\ \Rightarrow & g_{1}\left(z_{1}\right)⩽\frac{g_{1}\left(z_{0}\right)+\left|S\right|g_{2}\left(z_{0}\right)}{1-|S|}<0 \end{array}$$
which contradicts the fact that $g_{1}\left(z\right)⩾0$. Therefore, $g_{2}\left(z\right)⩽0$.4.Prove that $g_{1}\left(z\right)≢0$. We prove this case by contradiction. Suppose that $g_{1}\left(z\right)\equiv 0$. Based on the definition of h(k,z) and Equation (Equation 13), there exists $z_{0}\in Z$ such that
$$\min_{z\in Z}h\left(1,z\right)=\min_{z\in Z}(g_{1}\left(z\right)+g_{2}\left(z\right))=\min_{z\in Z}g_{2}\left(z\right)=g_{2}\left(z_{0}\right)=0.$$In other words,
$$g_{2}\left(z\right)⩾\min_{z\in Z}g_{2}\left(z\right)=g_{2}\left(z_{0}\right)=0.$$If $g_{2}\left(z\right)>0$, it contradicts the fact that $g_{2}\left(z\right)⩽0$; if $g_{2}\left(z\right)=0$, then $\min_{z\in W}h\left(k,z\right)=\theta \left(k\right)c$ that cannot be equal to 𝟙(k=|S|), contradicting the definition of h(k,z). Thus, $g_{1}\left(z\right)≢0$.
□

**Theorem 2.** 
*Let g(x,z) be the GRF of $x_{1}x_{2}\cdots x_{\left|S\right|}$ and h(k,z) be the auxiliary function of g(x,z). There exists $z_{0}\in W$ such that at most three integers $0⩽k_{1}<k_{2}<k_{3}⩽\left|S\right|$ satisfy $h(k_{1},z_{0})=0$, $h(k_{2},z_{0})=0$, $h\left(k_{3},z_{0}\right)=𝟙(k_{3}=\left|S|\right)$.*


**Proof.** According to the description in this theorem, $\exists z_{0}\in W$ is such that $h(k_{1},z_{0})=0$, $h(k_{2},z_{0})=0$, $h\left(k_{3},z_{0}\right)=𝟙(k_{3}=\left|S|\right)$, where three integers $0⩽k_{1}<k_{2}<k_{3}⩽\left|S\right|$. Based on the Equation (Equation 13), we have
(17)$$\left\{\begin{array}{cc} g_{1}\left(z_{0}\right)+k_{1}g_{2}\left(z_{0}\right)+\frac{k_{1}\left(k_{1}-1\right)}{2}c & =0,\\ g_{1}\left(z_{0}\right)+k_{2}g_{2}\left(z_{0}\right)+\frac{k_{2}\left(k_{2}-1\right)}{2}c & =0,\\ g_{1}\left(z_{0}\right)+k_{3}g_{2}\left(z_{0}\right)+\frac{k_{3}\left(k_{3}-1\right)}{2}c & =𝟙(k_{3}=|S|). \end{array}\right.$$If $k_{3}\neq \left|S\right|$, then c=0 which contradicts Theorem 1; if $k_{3}=\left|S\right|$, there exists a solution of Equation (Equation 17) such that $h(k_{1},z_{0})=0$, $h(k_{2},z_{0})=0$, $h\left(k_{3},z_{0}\right)=𝟙(k_{3}=\left|S|\right)=1$. Therefore, There exists $z_{0}\in W$ such that three integers $0⩽k_{1}<k_{2}<k_{3}⩽\left|S\right|$ satisfy $h(k_{1},z_{0})=0$, $h(k_{2},z_{0})=0$, $h\left(k_{3},z_{0}\right)=𝟙(k_{3}=\left|S|\right)$.Now we prove that $z_{0}\in W$ corresponds to at most three integers. We prove this by contradiction. Suppose that there exists $z_{0}\in W$ such that four integers $0⩽k_{1}<k_{2}<k_{3}<k_{4}⩽\left|S\right|$ satisfy $h(k_{1},z_{0})=0$, $h(k_{2},z_{0})=0$, $h(k_{3},z_{0})=0$ and $h\left(k_{4},z_{0}\right)=𝟙(k_{4}=\left|S|\right)$. Then, according to the Equation (Equation 13), we have
(18)$$\left\{\begin{array}{cc} g_{1}\left(z_{0}\right)+k_{1}g_{2}\left(z_{0}\right)+\frac{k_{1}\left(k_{1}-1\right)}{2}c & =0,\\ g_{1}\left(z_{0}\right)+k_{2}g_{2}\left(z_{0}\right)+\frac{k_{2}\left(k_{2}-1\right)}{2}c & =0,\\ g_{1}\left(z_{0}\right)+k_{3}g_{2}\left(z_{0}\right)+\frac{k_{3}\left(k_{3}-1\right)}{2}c & =0,\\ g_{1}\left(z_{0}\right)+k_{4}g_{2}\left(z_{0}\right)+\frac{k_{4}\left(k_{4}-1\right)}{2}c & =𝟙(k_{4}=|S|). \end{array}\right.$$To hold the equivalence of Equation (Equation 18), $k_{4}$ must be less than |S|, i.e., $k_{4}<\left|S\right|$. At this point, *c* must be zero, which contradicts Theorem 1. For the case that there are more than four integers, the proof is similar. □

**Corollary 1.** 
*Let g(x,z) be the GRF of $x_{1}x_{2}\cdots x_{\left|S\right|}$ and h(k,z) be the auxiliary function of g(x,z). If $z_{0}\in W$ is such that $h(k_{1},z_{0})=0$, $h(k_{2},z_{0})=0$, $h\left(k_{3},z_{0}\right)=𝟙(k_{3}=\left|S|\right)$, where three integers $0⩽k_{1}<k_{2}<k_{3}⩽\left|S\right|$, then $k_{3}=\left|S\right|$ and*

$$c=\frac{2}{\left(|S\right|-k_{1})(|S|-k_{2})}$$



**Proof.** Based on Theorem 2, to make Equation (Equation 17) hold and not contradict the fact that c>0, $k_{3}$ must be equal to |S| and the solution of Equation (Equation 17) is
$$g_{1}\left(z_{0}\right)=\frac{k_{1}k_{2}}{\left(|S\right|-k_{1})(|S|-k_{2})}$$
$$g_{2}\left(z_{0}\right)=\frac{1-k_{1}-k_{2}}{\left(|S\right|-k_{1})(|S|-k_{2})}$$
$$c=\frac{2}{\left(|S\right|-k_{1})(|S|-k_{2})}.$$□

**Theorem 3.** 
*Let g(x,z) be a GRF of $x_{1}x_{2}\cdots x_{\left|S\right|}$ and W be the value space of z. The minimum size of W is $\left\lceil \frac{\left|S\right|}{2}\right\rceil$.*


**Proof.** In order to obtain the minimum size of W, we should let the value $z_{0}\in W$ correspond to as many values of *k* as possible to satisfy $\min_{z\in W}h\left(k,z_{0}\right)=𝟙\left(k=|S|\right)$. To do this, we choose one value $z_{0}\in W$ such that $\min_{z\in W}h\left(k_{1},z_{0}\right)=0$, $\min_{z\in W}h\left(k_{2},z_{0}\right)=0$ and $\min_{z\in W}h\left(\left|S\right|,z_{0}\right)=1$, where $0⩽k_{1}<k_{2}<\left|S\right|$, based on Theorem 2. For $\forall z_{1}\in W$ and $z_{1}\neq z_{0}$, let $z_{1}$ correspond to two integers $0⩽j_{1}<j_{2}<\left|S\right|$, $j_{1}\neq k_{1}$ and $j_{2}\neq k_{2}$ such that $\min_{z\in W}h\left(j_{1},z_{1}\right)=0$ and $\min_{z\in W}h\left(j_{2},z_{1}\right)=0$ (or $\min_{z\in W}h\left(j_{1},z_{1}\right)=0$, $\min_{z\in W}h\left(j_{2},z_{1}\right)=0$ and $\min_{z\in W}h(|S|,z_{1})=1$). Therefore, for k∈{0,1,…,|S|}, we need at least $\left\lceil \frac{\left|S\right|}{2}\right\rceil$ values in W to ensure $\min_{z\in W}h\left(k,z\right)=𝟙\left(k=\left|S\right|\right)$. The theorem is proved. □

Apparently, the size of $W_{3}$ in Equation (Equation 16) is $\left\lceil \frac{\left|S\right|}{2}\right\rceil$ that matches the minimum size of W. Although Equations (Equation 14) and (Equation 15) do not satisfy the minimum size of W, they can not only derive some state-of-the-art order reduction methods shown in the next section, but also help to better analyze our framework shown in the experiments.

## 4. The Second Process of the Proposed Framework: Transformation from GRF to RF

This section proposes two transformations that convert the GRF g(x,z) to the RF f(x,y) in Section 4.1 and Section 4.2.

### 4.1. Substitution Transformation

The first is the substitution transformation, which defines z(y) and then substitutes *z* in g(x,z) with z(y):(19)$$g\left(x,z\right)\overset{z=z(y)}{\to }g\left(x,z\left(y\right)\right)=f\left(x,y\right).$$To ensure that f(x,y) is quadratic, z(y) must be linear and cover all values of *z*. Thus, we provide three forms:(20)$$\begin{array}{cc} \; & z\left(y\right)=\left(|W|-m\right)y_{1}+\sum_{i=2}^{m}y_{i},\;\frac{\left|W\right|}{2}⩽m⩽\left|W\right|-1 \end{array}$$(21)$$\begin{array}{cc} \; & z\left(y\right)=\left(|W|-2m+1\right)y_{1}+\sum_{i=2}^{m}2y_{i},\;\frac{\left|W\right|}{4}⩽m⩽\frac{\left|W\right|}{2} \end{array}$$(22)$$\begin{array}{cc} \; & z\left(y\right)=\sum_{i=1}^{m}2^{i-1}y_{i},\;m=\lceil \log_{2}\left(|W|)\right\rceil \end{array}$$Then, substituting *z* in Equations (Equation 15) and (Equation 16) with Equation (Equation 20) or (21) or (22), we obtain six novel order reduction methods. By inserting Equation (Equation 20) into Equation (Equation 15), the *substitution reduction type-1* (SR-1) method is defined as:(23)$$\begin{array}{cc} x_{1}x_{2}\cdots x_{\left|S\right|}= & \min_{y\in {\left\{0,1\right\}}^{m}}g\left(x,z\left(y\right)\right)=\min_{y\in {\left\{0,1\right\}}^{m}}{∥z\left(y\right)-x_{1}-\cdots -x_{\left|S\right|}∥}_{2}^{2}\\ = & \min_{y\in {\left\{0,1\right\}}^{m}}∥(|S|-m)y_{1}+\sum_{i=2}^{m}y_{i}-x_{1}-\cdots -x_{\left|S\right|}{∥}_{2}^{2},\;\frac{\left|S\right|}{2}⩽m⩽\left|S\right|-1 \end{array}$$Similarly, we substitute *z* in Equation (Equation 15) with Equation (21) and the *substitution reduction type-2* (SR-2) method is
(24)$$x_{1}x_{2}\cdots x_{\left|S\right|}=\min_{y\in {\left\{0,1\right\}}^{m}}∥(|S|-2m+1)y_{1}+\sum_{i=2}^{m}2y_{i}-x_{1}-\cdots -x_{\left|S\right|}{∥}_{2}^{2},\;\frac{\left|S\right|}{4}⩽m⩽\frac{\left|S\right|}{2}.$$Substituting *z* in Equation (Equation 15) with Equation (22), the *substitution reduction type-3* (SR-3) method is
(25)$$x_{1}x_{2}\cdots x_{\left|S\right|}=\min_{y\in {\left\{0,1\right\}}^{m}}∥\sum_{i=1}^{m}2^{i-1}y_{i}-x_{1}-\cdots -x_{\left|S\right|}{∥}_{2}^{2},\;m=\lceil \log_{2}(|S|)\rceil .$$Substituting *z* in Equation (Equation 16) with Equation (Equation 20), the *substitution reduction type-4* (SR-4) method is
(26)$$\begin{array}{cc} x_{1}x_{2}\cdots x_{\left|S\right|}=\min_{y\in {\left\{0,1\right\}}^{m}}\frac{1}{2} & (2\left(\left\lceil \frac{\left|S\right|}{2}\right\rceil -m\right)y_{1}+\sum_{i=2}^{d}2y_{i}-\sum_{i=1}^{m}x_{i})\cdot \\ \; & \left(2\left(\left\lceil \frac{\left|S\right|}{2}\right\rceil -m\right)y_{1}+\sum_{i=2}^{d}2y_{i}+{\left(-1\right)}^{\left|S\right|}-\sum_{i=1}^{\left|S\right|}x_{i}\right) \end{array}$$
where $\frac{\left\lceil \frac{\left|S\right|}{2}\right\rceil }{2}⩽m⩽\left\lceil \frac{\left|S\right|}{2}\right\rceil -1$. Substituting *z* in Equation (Equation 16) with Equation (21), the *substitution reduction type-5* (SR-5) method is
(27)$$\begin{array}{cc} x_{1}x_{2}\cdots x_{\left|S\right|}=\min_{y\in {\left\{0,1\right\}}^{m}}\frac{1}{2} & (2\left(\left\lceil \frac{\left|S\right|}{2}\right\rceil -2m+1\right)y_{1}+\sum_{i=2}^{m}4y_{i}-\sum_{i=1}^{\left|S\right|}x_{i})\cdot \\ \; & \left(2\left(\left\lceil \frac{\left|S\right|}{2}\right\rceil -2m+1\right)y_{1}+\sum_{i=2}^{m}4y_{i}+{\left(-1\right)}^{\left|S\right|}-\sum_{i=1}^{\left|S\right|}x_{i}\right) \end{array}$$
where $\frac{\left\lceil \frac{\left|S\right|}{2}\right\rceil }{4}⩽m⩽\frac{\left\lceil \frac{\left|S\right|}{2}\right\rceil }{2}$. Substituting *z* in Equation (Equation 16) with Equation (22), the *substitution reduction type-6* (SR-6) method is
(28)$$x_{1}x_{2}\cdots x_{\left|S\right|}=\min_{y\in {\left\{0,1\right\}}^{m}}\frac{1}{2}(\sum_{i=1}^{m}2^{i}y_{i}-\sum_{i=1}^{\left|S\right|}x_{i})(\sum_{i=1}^{m}2^{i}y_{i}+{\left(-1\right)}^{\left|S\right|}-\sum_{i=1}^{\left|S\right|}x_{i})$$
where $m=\lceil \log_{2}\left(\left\lceil \frac{\left|S\right|}{2}\right\rceil \right)\rceil$.

The RFs transformed from Equations (Equation 20)–(22) is simpler and clearer than some state-of-the-art reduction methods. For example, substituting *z* in Equation (Equation 15) with
$$z\left(y\right)=\sum_{i=0}^{m-1}2^{i}y_{i}-2^{m}+\left|S\right|+\frac{1-{\left(-1\right)}^{\left|S\right|}}{2}$$
where $m=\lceil \log_{1}\left(|S|)\right\rceil$, we obtain LogR-1 Equation (Equation 8) that has an extra $-2^{m}+\left|S\right|$ and thus is more complex than SR-3 Equation (Equation 25). Substituting *z* in Equation (Equation 16) with
$$z\left(y\right)=\sum_{i=1}^{m}2^{i-1}y_{i}-2^{m}+\frac{\left|S\right|}{2}+\frac{1-{\left(-1\right)}^{\left|S\right|}}{4}$$
where $m=\lceil \log_{1}\left(|S|)\right\rceil -1$, we obtain LogR-2 Equation (Equation 9) that has an extra $-2^{m}+\frac{\left|S\right|}{2}$ and thus is more complex than SR-6 Equation (Equation 28). Note that LinR can be viewed as substituting *z* in Equation (Equation 16) with
$$z\left(y\right)=\frac{1}{2}\left(\left|S\right|-2m\right)y_{1}+\sum_{i=2}^{m}y_{i}+\frac{1-{\left(-1\right)}^{\left|S\right|}}{4}$$
where |S|/4⩽m⩽|S|/2.

### 4.2. Minimum Transformation

Unlike Equation (Equation 19), the minimum transformation utilizes the minimum operation to obtain f(x,y):(29)$$g\left(x,z\right)\overset{z=z(y)}{\to }g\left(x,z\left(y\right)\right)=\min_{y\in {\left\{0,1\right\}}^{d}}f\left(x,y\right).$$Compared with the proposed substitution transformation, the minimum transformation is capable of avoiding the exponential coefficients in Equation (22) and transforming Equation (Equation 14) into an RF. There are two specific forms.

First, suppose a vector of auxiliary variables: $y=\left(y_{1},\ldots ,y_{m}\right)\in {\left\{0,1\right\}}^{m}$ where m=|W|−1. We define
(30)$$z\left(y\right)=y_{1}+y_{1}y_{2}+\cdots +y_{1}y_{2}\cdots y_{m}.$$By inserting Equation (Equation 30) into Equation (Equation 14), we have
$$\begin{array}{cc} g(x,z(y))= & 𝟙\left(z\left(y\right)=|S|\right)+z^{2}\left(y\right)-2z\left(y\right)\sum_{i=1}^{\left|S\right|}x_{i}+\left(\sum_{i=1}^{\left|S\right|}x_{i}\right){}^{2}\\ = & 𝟙(\sum_{i=1}^{\left|S\right|}\prod_{j=1}^{i}y_{j}=\left|S|\right)+\sum_{i=1}^{\left|S\right|}\left(2i-1-2\sum_{i=1}^{\left|S\right|}x_{i}\right)\prod_{j=1}^{i}y_{j}+\left(\sum_{i=1}^{\left|S\right|}x_{i}\right){}^{2}\\ = & \min_{y\in {\left\{0,1\right\}}^{\left|S\right|}}y_{\left|S\right|}+\sum_{i=1}^{\left|S\right|}\left(2i-1-2\sum_{i=1}^{\left|S\right|}x_{i}\right)y_{i}+\left(\sum_{i=1}^{\left|S\right|}x_{i}\right){}^{2}\\ = & \min_{y\in {\left\{0,1\right\}}^{\left|S\right|}}f\left(x,y\right). \end{array}$$Thus, the *minimum reduction type-1* (MR-1) method is
(31)$$x_{1}x_{2}\cdots x_{\left|S\right|}=\min_{z\in W}g\left(x,z\right)=\min_{y\in {\left\{0,1\right\}}^{\left|S\right|}}y_{\left|S\right|}+\sum_{i=1}^{\left|S\right|}\left(2i-1-2\sum_{i=1}^{\left|S\right|}x_{i}\right)y_{i}+(\sum_{i=1}^{\left|S\right|}x_{i}{)}^{2}.$$Similarly, for the Equation (Equation 15) case, we have the *minimum reduction type-2* (MR-2) method:(32)$$x_{1}x_{2}\cdots x_{\left|S\right|}=\min_{y\in {\left\{0,1\right\}}^{|S|-1}}\sum_{i=1}^{|S|-1}\left(2i-1-2\sum_{i=1}^{\left|S\right|}x_{i}\right)y_{i}+(\sum_{i=1}^{\left|S\right|}x_{i}{)}^{2}.$$For the Equation (Equation 16) case, we have the *minimum reduction type-3* (MR-3) method:(33)$$\begin{array}{cc} x_{1}x_{2}\cdots x_{\left|S\right|}=\min_{y\in {\left\{0,1\right\}}^{\left\lfloor \frac{\left|S\right|-1}{2}\right\rfloor }} & \sum_{i=1}^{\left\lfloor \frac{\left|S\right|-1}{2}\right\rfloor }(4i-2+{\left(-1\right)}^{\left|S\right|}-2\sum_{i=1}^{\left|S\right|}x_{i})y_{i}\\ \; & +\sum_{1⩽i<j⩽\left|S\right|}x_{i}x_{j}+\frac{1-{\left(-1\right)}^{\left|S\right|}}{2}\sum_{i=1}^{\left|S\right|}x_{i}. \end{array}$$

Second, suppose a vector of auxiliary variables: $y=\left(y_{1},\ldots ,y_{m},y_{1}^{\prime },\ldots ,y_{m}^{\prime }\right)\in {\left\{0,1\right\}}^{2m}$ where $m=\lceil \sqrt{\left|W\right|}\rceil$. We define
(34)$$z\left(y\right)=m\sum_{i=1}^{m}\left(i-1\right)\cdot y_{i}+\sum_{j=1}^{m}\left(j-1\right)\cdot y_{j}^{\prime }.$$To ensure the equivalence of Equation (Equation 29), we add a penalty term in f(x,y), which is:(35)$$x_{1}x_{2}\cdots x_{\left|S\right|}=\min_{y\in {\left\{0,1\right\}}^{2m}}f\left(x,y\right)=\min_{y\in {\left\{0,1\right\}}^{2m}}g\left(x,z\left(y\right)\right)+\lambda ((1-\sum_{i=1}^{m}y_{i}{)}^{2}+(1-\sum_{j=1}^{m}y_{j}^{\prime }{)}^{2})$$
where λ is a large number. For Equation (Equation 14), $𝟙\left(z\left(y\right)=|S|\right)=y_{i_{0}}y_{i_{0}}^{\prime }$ where $y_{i_{0}}$ and $y_{i_{0}}^{\prime }$ satisfy $m\left(i_{0}-1\right)\cdot y_{i_{0}}+\left(j_{0}-1\right)\cdot y_{j_{0}}^{\prime }=\left|S\right|$. By substituting *z* and 𝟙(z=|S|) in Equation (Equation 14) with z(y) and 𝟙(z(y)=|S|), the SqrtR Equation (Equation 11) is obtained. Similarly, substituting *z* in Equation (Equation 15) with Equation (Equation 34), we obtain the *minimum reduction type-4* (MR-4) method:(36)$$\begin{array}{cc} x_{1}x_{2}\cdots x_{\left|S\right|}=\min_{y\in {\left\{0,1\right\}}^{2m}} & ∥m\sum_{i=1}^{m}\left(i-1\right)\cdot y_{i}+\sum_{j=1}^{m}\left(j-1\right)\cdot y_{j}^{\prime }-\sum_{k=1}^{\left|S\right|}x_{k}{∥}_{2}^{2}\\ \; & +\lambda ((1-\sum_{i=1}^{m}y_{i}{)}^{2}+(1-\sum_{j=1}^{m}y_{j}^{\prime }{)}^{2}) \end{array}$$
where $m=\lceil \sqrt{\left|S\right|}\rceil$. Finally, substituting *z* in Equation (Equation 16) with with Equation (Equation 34), we obtain the *minimum reduction type-5* (MR-5) method:(37)$$\begin{array}{cc} x_{1}x_{2}\cdots x_{\left|S\right|}=\min_{y\in {\left\{0,1\right\}}^{2m}}\; & \frac{1}{2}(2m\sum_{i=1}^{m}\left(i-1\right)\cdot y_{i}+\sum_{j=1}^{m}2\left(j-1\right)\cdot y_{j}^{\prime }-\sum_{k=1}^{\left|S\right|}x_{k})\\ \; & \cdot (2m\sum_{i=1}^{m}\left(i-1\right)\cdot y_{i}+\sum_{j=1}^{m}2\left(j-1\right)\cdot y_{j}^{\prime }+{\left(-1\right)}^{\left|S\right|}-\sum_{k=1}^{\left|S\right|}x_{k})\\ \; & +\lambda ((1-\sum_{i=1}^{m}y_{i}{)}^{2}+(1-\sum_{j=1}^{m}y_{j}^{\prime }{)}^{2}) \end{array}$$
where $m=\lceil \sqrt{\left\lceil \frac{\left|S\right|}{2}\right\rceil }\;\rceil$.

## 5. Experiments and Discussions

In this section, we compare the proposed method (GR-1, GR-2, GR-3, SR-1, SR-2, SR-3, SR-4, SR-5, SR-6, MR-1, MR-2, MR-3, MR-4 and MR-5) with state-of-the-art order reduction methods (HOCR, LogR-1, LogR-2, LinR and SqrtR) on synthetic data and image denoising. Table 1 summarizes the order reduction methods developed in this paper. To ensure the fairness of the comparison, we utilize the classical algorithm TRW-S [44,45] to minimize the RMRFs reduced by these order reduction methods, and conduct the comparison experiments with the same hardware and software environment (AMD Ryzen 5800U, RAM 16GB, Nvidia RTX 3050 and MATLAB 2022a).

### 5.1. Synthetic Data Experiments

In the first experiment, we follow the approach in the paper [46] to generate a series of synthetic HoMRFs, which have the following form:(38)$$E\left(x_{1},x_{2},\cdots ,x_{n}\right)=\sum_{i=1}^{n}c_{i}x_{i}+\sum_{S\subseteq [n],|S|>2}E_{S}\left(x_{S_{1}},x_{S_{2}}\ldots ,x_{S_{\left|S\right|}}\right)$$
where $x_{1},x_{2},\ldots ,x_{n}\in \left\{0,1\right\}$, $c_{i}\in R$ is the coefficient, and $E_{S}:{\left\{0,1\right\}}^{\left|S\right|}\to R$. We synthesize HoMRfs of order three to seven. For each order, we generate 500 HoMRFs with n=50, 500 HoMRFs with n=200, and 10 HoMRFs with n=1000. The variables $x_{1},x_{2},\ldots ,x_{n}$ are randomly sampled from a uniform distribution, and the coefficient $c_{i}$ and the values of $E_{S}$ are randomly sampled from a standard Gaussian distribution. The energy results of each order achieved by different order reduction methods are shown in Table 2, Table 3 and Table 4, respectively. Note that for SqrtR, MR-4 and MR-5, λ=5.

First, from Table 2, Table 3 and Table 4, it can be seen that for the third-order HoMRFs, the best energy performance is obtained by HOCR, LinR and MR-3; for the forth-order HoMRFs, the best energy performance is obtained by GR-3; for the fifth-order to seventh-order HoMRFs, the best energy performance is obtained by HOCR and MR-3. As shown in Equations (Equation 7), (Equation 16) and (Equation 33), these methods with the best performance all have fewer number of auxiliary variables and smaller coefficients, which is strongly supported by the research [18,21,22,46]. Although the performance of HOCR is identical to that of MR-3, designing HOCR is more difficult than MR-3. Since HOCR is designed by summarizing the RF f(x,y) of the third-order to seventh-order positive monomials [1,2], it is heuristic. Therefore, it must be fortunate enough to heuristically find the required RFs from a large number of RFs. As for LinR, it is designed by a complex mathematical theory in the references [21,22]. In contrast, our design framework explicitly gives two processes, each of which is much more rigorous than HOCR [1,2] and simpler than LinR [21,22].

Moreover, as we discussed in Section 4.2, the state-of-the-art order reduction methods LogR-1 and LogR-2 [21,22] can be derived from our design framework. We also present two methods SR-3 and SR-6 that are more concise forms of LogR-1 and LogR-2. For the energy performance, SR-3 and SR-6 outperform LogR-1 and LogR-2, respectively, shown in Table 2, Table 3 and Table 4. It shows that the proposed framework produces superior order reduction methods than state-of-the-art ones.

Third, to validate the importance of the proposed theorems in Section 3.2, we compare the energy performance of GR-1 Equation (Equation 14), GR-2 Equation (Equation 15) and GR-3 Equation (Equation 16). As seen in Table 2, Table 3 and Table 4, while GR-3 achieves the best energy performance on the fourth-order HoMRFs and the second best energy performance on the fifth-order to seventh-order HoMRFs, GR-1 and GR-2 perform poorly. This result shows that GR-3 is more preferable than GR-1 and GR-2. Furthermore, the RFs transformed from GR-3 are also more preferable than those from GR-1 and GR-2: SR-4, SR-5 and SR-6 are superior to SR-1, SR-2 and SR-3 in energy performance, respectively; MR-3 outperforms MR-1 and MR-2, and MR-5 outperforms SqrtR and MR-4. All these results indicate the importance of GR-3 and demonstrate that the size of value space |W| does significantly affect the efficiency of the reduction method.

Finally, we compare the energy performance of substitution transformations Equations (Equation 20)–(22) and minimum transformations Equations (Equation 30) and (Equation 34) in Section 3.2. As shown in Table 2, Table 3 and Table 4, SR-4, SR-5 and SR-6 have exactly the same energy performance on the third-order and fourth-order HoMRFs, since their formulas are the same in lower order HoMRFs. For the fifth-order, sixth-order and seventh-order HoMRFs, SR-4, SR-5 and SR-6 have different energy performance, and the performance of SR-5 is superior to that of SR-4 and SR-6. It demonstrates that for the GR-3, the substitution transformation Equation (21) is better than other transformations Equations (Equation 20) and (22). Similarly, it can be seen from SR-1, SR-2 and SR-3 that no substitution transformations Equations (Equation 20)–(22) has obvious advantages for GR-2. Furthermore, MR-1 and MR-3 show the better energy than SqrtR and MR-5 respectively in any case, and MR-2 shows the better energy than MR-4 in most cases. This result reveals that with the same GRF, the minimum transformation Equation (Equation 30) is more preferable than the minimum transformation Equation (Equation 34). We can conclude that no matter what transformation g(x,z) chooses, the fewer auxiliary variables and smaller coefficients lead to superior results.

### 5.2. Image Denoising

In this section, we conduct the second experiment on the benchmark [1,2], *Fields of Expert* (FoE) [47,48], which is a widely used HoMRF model for image denoising. FoE represents the prior probability of an image distribution as:$$E\left(x_{1},\ldots ,x_{n}\right)=\sum_{i=1}^{n}\frac{{\left(a_{i}-x_{i}\right)}^{2}}{2\sigma^{2}}+\sum_{i=1}^{n}\alpha_{i}\log\left[1+\frac{1}{2}\left(A_{i}*x_{i}\right)\right]$$
where *n* is the number of pixels of the final image, $a_{i}$ is the value of the *i*-th pixel in the noise image, and $A_{i}*x_{i}$ denotes the result of convolving the patch at pixel *i* with filter $A_{i}$. The parameters $A_{i}$ and $\alpha_{i}$ are learned from a database of natural images [47,48]. For more details, please refer to [1,2,47,48]. In the benchmark [1,2], 10 images from the widely known Berkeley Segmentation Database (BSDS500) [47,48,49] are selected. Gaussian noise with a standard deviation σ is added to each image. For efficiency, we shrink the image size by half. To make a fair comparison, we set the same starting point and maximum number of 100 iterations for order reduction methods.

We utilize the energy and the *peak signal to noise ratio* (PSNR) and the calculation time to compare the performance of different reduction methods. The results are shown in Table 5. We also plot the curves of the energy and PSNR with iterations shown in Figure 2. A visual example of image denoising is illustrated in Figure 3. Note that for the FoE model, HOCR and LinR are the same. Thus, they have the same performance in terms of energy and PSNR and differ only slightly in terms of calculation time. The same is true for SR-4, SR-5 and SR-6. For SqrtR, MR-4 and MR-5, λ=5.

First, as shown in Table 5, the best performance of the energy, PSNR and calculation time is obtained by LinR, MR-5 and MR-3, respectively. For the denoised images with σ=20, MR-3 has 1.68% higher energy than LinR and 0.89% lower PSNR than MR-5. However, the calculation time of MR-3 is 8.73% and 55.31% lower than that of LinR and MR-5, respectively. Thus, MR-3 balances energy, PSNR and calculation time and achieves the overall best performance. Similarly, for the denoised images with σ={15,25}, MR-3 also obtains the overall best performance. As discussed in the synthetic data experiment, MR-3 also has the advantage of lower design difficulty, which fully demonstrates the superiority of the proposed framework.

Moreover, as we discussed in Section 4.2, the proposed SR-3 and SR-6 are simpler and clearer than LogR-1 and LogR-2. It can be seen from the Figure 2a and Table 5 that SR-3 outperforms LogR-1 in energy, PSNR and calculation time, while SR-6 outperforms LogR-2 in calculation time. The results indicate that the proposed framework can improve the performance of some state-of-the-art order reduction methods.

Third, we compare different GR-1 Equation (Equation 14), GR-2 Equation (Equation 15) and GR-3 Equation (Equation 16) in Section 3.1. As shown in Figure 2b and Table 5, GR-3 has better performance than GR-1 and GR-2 in terms of energy, PSNR and calculation time. In Figure 3, GR-1 and GR-2 keep more noise, while GR3 preserves the important edge information while removing noise, which demonstrates the superiority of Equation (Equation 16). Similarly, MR-3 outperforms MR-1 and MR-2, and MR-5 outperforms SqrtR and MR-4, shown in Figure 2d and Figure 3. This shows that the RF f(x,y) transformed from Equation (Equation 16) is superior to that from Equations (Equation 14) and (Equation 15). These results indicate that |W| significantly affect the efficiency of the reduction method and that proving the theorems in Section 2.2 is important.

Forth, we compare different transformations Equations (Equation 20)–(22), (Equation 30) and (Equation 34) in Section 3.2. As shown in Figure 2c and Table 5, SR-2 performs best among SR-1 and SR-3 in energy, PSNR and calculation time. The visual results in Figure 3 also show that SR-2 outperforms SR-1 and SR-3, which demonstrates that Equation (21) is more preferable than Equations (Equation 20) and (22). Similarly, MR-1 outperforms SqrtR in energy, PSNR and calculation time; MR-2 and MR-3 outperform MR-4 and MR-5, respectively, in energy and calculation time. This result demonstrates the significant advantage of Equation (Equation 30) over Equation (Equation 34) in energy and calculation time, while for PSNR, Equation (Equation 34) is slightly better than Equation (Equation 30).

Moreover, we analyze the stability and convergence of different order reduction methods. To numerically investigate the stability of order reduction methods, the input images in the benchmark [1,2] are added with Gaussian noise of different standard deviations σ={15,20,25}. The results are shown in Table 5. As it can be seen, different order reduction methods show almost the same hierarchy of denoising performances for σ={15,20,25}, which demonstrates that the order reduction methods exhibit a very robust response against the standard deviation changes. In Figure 4, to show the convergence of different order reduction methods, we plot the energy changes obtained by different order reduction methods for all images in the benchmark [1,2]. The experimental results show that all order reduction methods converge for all images in the benchmark [1,2]. Among these methods, the curves of LogR-1, SqrtR and SR-1 are not smooth enough, indicating that they are more susceptible to the influence of fusion move [1,2], which is a standard technique that converts Markov random field with any multi-labels into a binary Markov random field. Moreover, most of the proposed order reduction methods (GR-1, GR-2, GR-3, SR-2, SR-3, SR-4, SR-5, SR-6, MR-1, MR-2, MR-3, MR-4 and MR-5) converge smoothly to their respective energies, which illustrates the superiority of the proposed design framework.

Finally, we summarize two important conclusions that are helpful for designing effective reduction methods. First, the fewer the auxiliary variables y in f(x,y), the better the experimental performance. For example, as shown in Figure 2 and Table 5, LinR, LogR-2 and MR-3 with the least y in f(x,y) have excellent performance. Second, the smaller coefficients in f(x,y) are also beneficial to the experimental performance, especially the calculation time. For example, MR-3 has the fastest calculation time since it has the fewest coefficients y in f(x,y).

## 6. Conclusions

In this paper, we propose a novel framework that significantly reduces the design difficulty for order reduction methods. The framework generalizes previous order reduction methods and proposes a novel GRF, which is then transformed into more RFs. The experimental results validate the superiority of the proposed design framework. There are several interesting directions for future work. First, although the proposed order reduction methods are able to reduce HoMRF with any order, minimizing the RMRF reduced from HoMRF would fail when the order of HoMRF is particularly high. Therefore, the efficiency of order reduction methods is the priority for future research. Second, there is a few of applications for order reduction methods. We plan to introduce the order reduction method into more fields. 

## Figures and Tables

**Figure 1 entropy-25-00535-f001:**
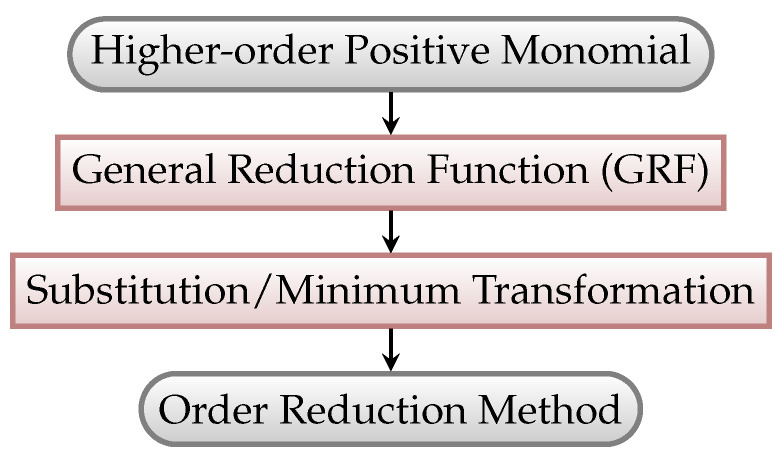
Graphical flowchart of the order reduction design framework for the higher-order positive monomial.

**Figure 2 entropy-25-00535-f002:**
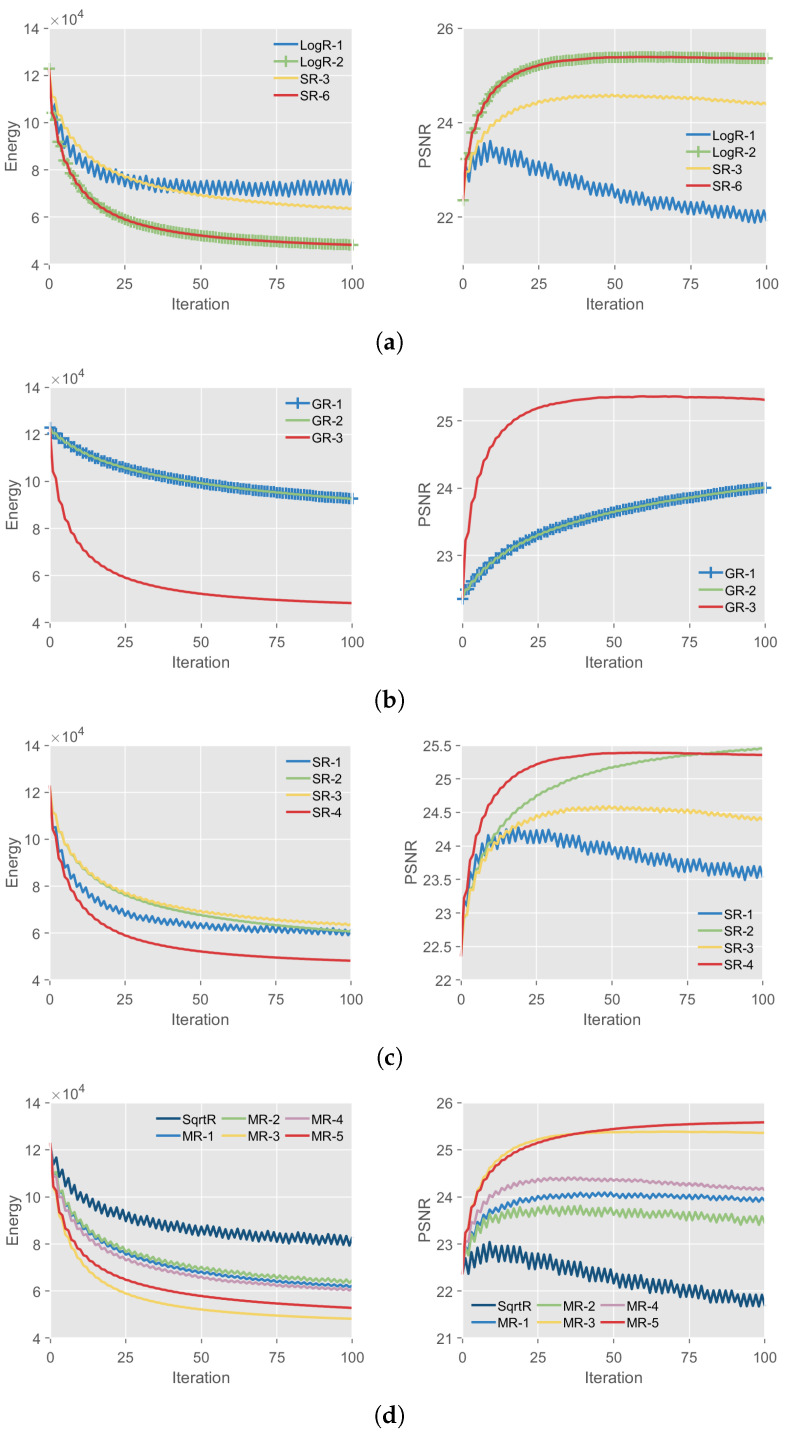
Energy and PSNR with iterations on the benchmark [1,2], averaged over the denoising results. (**a**) Comparison of order reduction methods LogR-1, LogR-2, SR-3 and SR-6. (**b**) Comparison of order reduction methods GR-1, GR-2 and GR-3. (**c**) Comparison of order reduction methods SR-1, SR-2, SR-3, SR-4, SR-5 and SR-6. Note that SR-4, SR-5 and SR-6 are identical for the FoE model. (**d**) Comparison of order reduction methods SqrtR, MR-1, MR-2, MR-3, MR-4 and MR-5.

**Figure 3 entropy-25-00535-f003:**
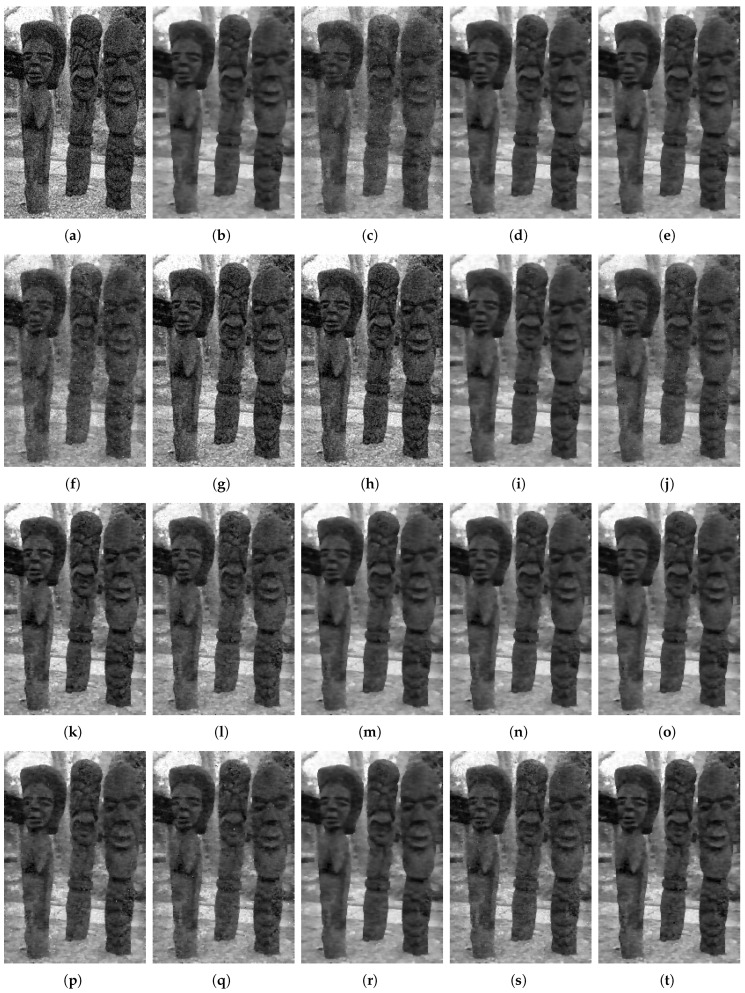
Visual example of image denoising. (**b**–**f**) Denoising results achieved by state-of-the-art order reduction methods. (**g**–**t**) Denoising results achieved by the proposed order reduction methods. (**a**) Noise Image. (**b**) HOCR Equation (Equation 7). (**c**) LogN-1 Equation (Equation 8). (**d**) LogN-2 Equation (Equation 9). (**e**) LinR Equation (Equation 10). (**f**) SqrtR Equation (Equation 11). (**g**) GR-1 Equation (Equation 14). (**h**) GR-2 Equation (Equation 15). (**i**) GR-3 Equation (Equation 16). (**j**) SR-1 Equation (Equation 23). (**k**) SR-2 Equation (Equation 24). (**l**) SR-3 Equation (Equation 25). (**m**) SR-4 Equation (Equation 26). (**n**) SR-5 Equation (Equation 27). (**o**) SR-6 Equation (Equation 28). (**p**) MR-1 Equation (Equation 31). (**q**) MR-2 Equation (Equation 32). (**r**) MR-3 Equation (Equation 33). (**s**) MR-4 Equation (Equation 36). (**t**) MR-5 Equation (Equation 37).

**Figure 4 entropy-25-00535-f004:**
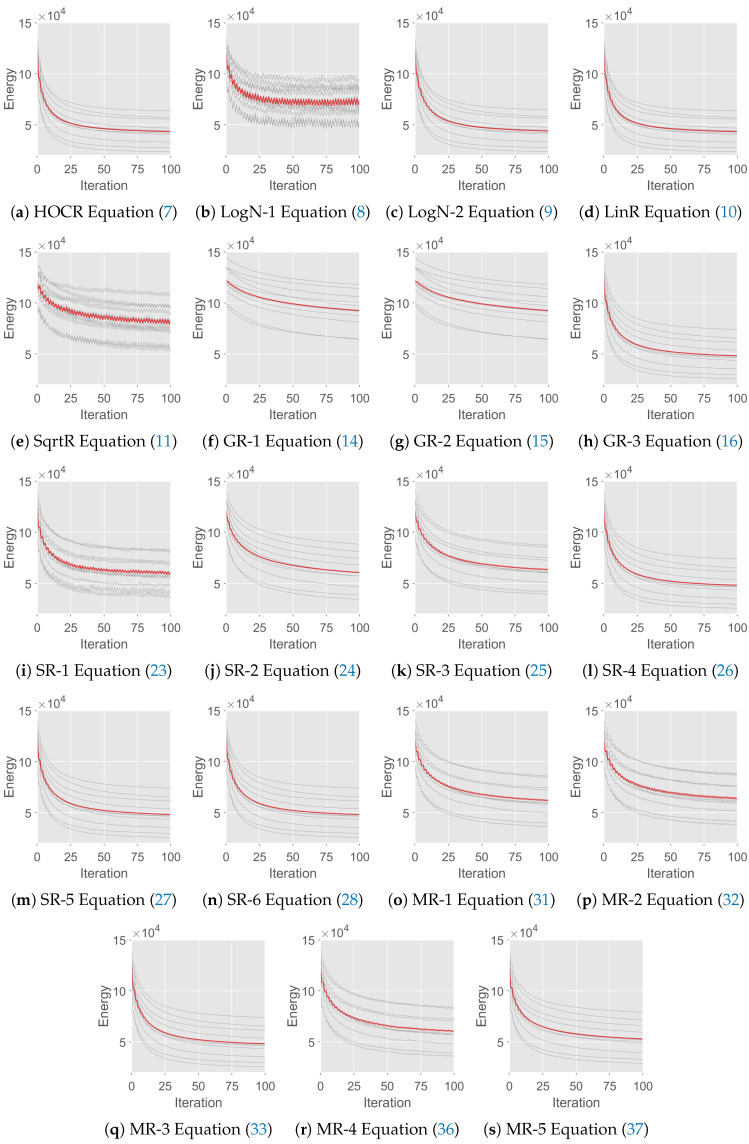
Energy with iterations achieved by different order reduction methods. In each subplot, every gray curve indicates the energy of one image in the benchmark [1,2], and the red curve indicates the average energy of all images in the benchmark [1,2].

**Table 1 entropy-25-00535-t001:** Comparison of the proposed 14 order reduction methods for the higher-order positive monomial $x_{1}x_{2}\cdots x_{\left|S\right|}$.

Methods	GRF	Transformation	Number of Auxiliary Variables	Type of Auxiliary Variables
GR-1 Equation (Equation 14)	(Equation 14)	None	m=1	Integral
GR-2 Equation (Equation 15)	(Equation 15)	None	m=1	Integral
GR-3 Equation (Equation 16)	(Equation 16)	None	m=1	Integral
SR-1 Equation (Equation 23)	(Equation 15)	(Equation 20)	$\frac{\left|S\right|}{2}⩽m⩽\left|S\right|-1$	Binary
SR-2 Equation (Equation 24)	(Equation 15)	(21)	$\frac{\left|S\right|}{4}⩽m⩽\frac{\left|S\right|}{2}$	Binary
SR-3 Equation (Equation 25)	(Equation 15)	(22)	$m=\lceil \log_{2}\left(|S|)\right\rceil$	Binary
SR-4 Equation (Equation 26)	(Equation 16)	(Equation 20)	$\frac{\left\lceil \frac{\left|S\right|}{2}\right\rceil }{2}⩽m⩽\left\lceil \frac{\left|S\right|}{2}\right\rceil -1$	Binary
SR-5 Equation (Equation 27)	(Equation 16)	(21)	$\frac{\left\lceil \frac{\left|S\right|}{2}\right\rceil }{4}⩽m⩽\frac{\left\lceil \frac{\left|S\right|}{2}\right\rceil }{2}$	Binary
SR-6 Equation (Equation 28)	(Equation 16)	(22)	$m=\lceil \log_{2}\left(\left\lceil \frac{\left|S\right|}{2}\right\rceil \right)\rceil$	Binary
MR-1 Equation (Equation 31)	(Equation 14)	(Equation 30)	m=|S|	Binary
MR-2 Equation (Equation 32)	(Equation 15)	(Equation 30)	m=|S|−1	Binary
MR-3 Equation (Equation 33)	(Equation 16)	(Equation 30)	$m=\lfloor \frac{\left|S\right|-1}{2}\rfloor$	Binary
MR-4 Equation (Equation 36)	(Equation 15)	(Equation 34)	$m=\lceil \sqrt{\left|S\right|}\rceil$	Binary
MR-5 Equation (Equation 37)	(Equation 16)	(Equation 34)	$m=\lceil \sqrt{\left\lceil \frac{\left|S\right|}{2}\right\rceil }\;\rceil$	Binary

**Table 2 entropy-25-00535-t002:** Comparison of energy (×10) averaged over 500 HoMRFs with 50 variables, achieved by different order reduction methods. We highlight the best performance in bold.

Methods	3rd-Order	4th-Order	5th-Order	6th-Order	7th-Order
HOCR Equation (Equation 7)	**−9.98**	−10.03	**−13.04**	**−15.72**	**−19.22**
LogR-1 Equation (Equation 8)	−6.22	−3.72	−3.21	−6.53	−7.49
LogR-2 Equation (Equation 9)	−8.82	−10.03	−10.12	−8.98	−9.99
LinR Equation (Equation 10)	**−9.98**	−10.03	−10.10	−10.30	−11.89
SqrtR Equation (Equation 11)	−2.89	−3.46	−5.08	−6.82	−8.45
GR-1 Equation (Equation 14)	−0.59	−0.16	−0.06	−0.02	−0.01
GR-2 Equation (Equation 15)	−0.59	−0.16	−0.06	−0.02	−0.01
GR-3 Equation (Equation 16)	−7.92	**−10.21**	−10.99	−15.41	−17.17
SR-1 Equation (Equation 23)	−6.19	−5.38	−6.16	−6.73	−7.56
SR-2 Equation (Equation 24)	−4.86	−4.85	−5.76	−7.15	−6.94
SR-3 Equation (Equation 25)	−4.19	−3.72	−7.91	−8.30	−8.68
SR-4 Equation (Equation 26)	−8.82	−10.03	−7.72	−10.3	−7.95
SR-5 Equation (Equation 27)	−8.82	−10.03	−10.02	−14.35	−11.79
SR-6 Equation (Equation 28)	−8.82	−10.03	−6.73	−10.22	−9.99
MR-1 Equation (Equation 31)	−6.01	−6.35	−8.89	−12.18	−14.56
MR-2 Equation (Equation 32)	−5.95	−6.53	−8.97	−11.75	−14.46
MR-3 Equation (Equation 33)	**−9.98**	−10.03	**−13.04**	**−15.72**	**−19.22**
MR-4 Equation (Equation 36)	−5.19	−4.22	−7.84	−9.34	−8.49
MR-5 Equation (Equation 37)	−6.68	−8.93	−9.08	−11.59	−12.01

The order reduction methods below the solid line are designed by our framework.

**Table 3 entropy-25-00535-t003:** Comparison of energy ($\times 10^{2}$) averaged over 500 HoMRFs with 200 variables, achieved by different order reduction methods. We highlight the best performance in bold.

Methods	3rd-Order	4th-Order	5th-Order	6th-Order	7th-Order
HOCR Equation (Equation 7)	**−4.11**	−4.00	**−4.86**	**−5.74**	**−6.88**
LogR-1 Equation (Equation 8)	−2.61	−1.44	−1.28	−2.68	−3.28
LogR-2 Equation (Equation 9)	−3.52	−4.00	−4.18	−3.69	−4.10
LinR Equation (Equation 10)	**−4.11**	−4.00	−3.81	−3.75	−4.37
SqrtR Equation (Equation 11)	−1.12	−1.39	−2.20	−3.09	−3.96
GR-1 Equation (Equation 14)	−0.18	−0.05	−0.01	−0.01	0.00
GR-2 Equation (Equation 15)	−0.18	−0.05	−0.01	−0.01	0.00
GR-3 Equation (Equation 16)	−3.24	**−4.05**	−4.13	−5.52	−6.15
SR-1 Equation (Equation 23)	−2.57	−2.11	−2.22	−2.24	−2.36
SR-2 Equation (Equation 24)	−1.91	−1.86	−2.21	−2.67	−2.66
SR-3 Equation (Equation 25)	−1.67	−1.44	−3.26	−3.55	−3.78
SR-4 Equation (Equation 26)	−3.52	−4.00	−2.76	−3.75	−2.51
SR-5 Equation (Equation 27)	−3.52	−4.00	−4.07	−5.41	−4.82
SR-6 Equation (Equation 28)	−3.52	−4.00	−2.46	−3.81	−4.11
MR-1 Equation (Equation 31)	−2.36	−2.36	−3.24	−4.14	−5.21
MR-2 Equation (Equation 32)	−2.34	−2.41	−3.21	−4.00	−5.00
MR-3 Equation (Equation 33)	**−4.11**	−4.00	**−4.86**	**−5.74**	**−6.88**
MR-4 Equation (Equation 36)	−2.10	−1.62	−3.30	−3.86	−3.47
MR-5 Equation (Equation 37)	−2.75	−3.54	−3.69	−4.64	−5.05

The order reduction methods below the solid line are designed by our framework.

**Table 4 entropy-25-00535-t004:** Comparison of energy ($\times 10^{3}$) averaged over 10 HoMRFs with 1000 variables, achieved by different order reduction methods. We highlight the best performance in bold.

Methods	3rd-Order	4th-Order	5th-Order	6th-Order	7th-Order
HOCR Equation (Equation 7)	**−1.97**	−1.92	**−2.21**	**−2.66**	**−3.15**
LogR-1 Equation (Equation 8)	−1.25	−0.66	−0.60	−1.30	−1.60
LogR-2 Equation (Equation 9)	−1.65	−1.92	−2.02	−1.72	−1.86
LinR Equation (Equation 10)	**−1.97**	−1.92	−1.80	−1.64	−1.95
SqrtR Equation (Equation 11)	−0.51	−0.64	−1.01	−1.40	−1.84
GR-1 Equation (Equation 14)	−0.07	−0.02	−0.01	0.00	0.00
GR-2 Equation (Equation 15)	−0.07	−0.02	−0.01	0.00	0.00
GR-3 Equation (Equation 16)	−1.57	**−1.94**	−2.00	−2.57	−2.91
SR-1 Equation (Equation 23)	−1.22	−0.97	−0.96	−0.99	−0.93
SR-2 Equation (Equation 24)	−0.86	−0.85	−1.06	−1.27	−1.21
SR-3 Equation (Equation 25)	−0.74	−0.66	−1.61	−1.72	−1.85
SR-4 Equation (Equation 26)	−1.65	−1.92	−1.29	−1.64	−1.01
SR-5 Equation (Equation 27)	−1.65	−1.92	−1.99	−2.54	−2.30
SR-6 Equation (Equation 28)	−1.65	−1.92	−1.12	−1.75	−1.86
MR-1 Equation (Equation 31)	−1.09	−1.08	−1.45	−1.82	−2.20
MR-2 Equation (Equation 32)	−1.09	−1.06	−1.45	−1.79	−2.17
MR-3 Equation (Equation 33)	**−1.97**	−1.92	**−2.21**	**−2.66**	**−3.15**
MR-4 Equation (Equation 36)	−0.97	−0.73	−1.53	−1.80	−1.46
MR-5 Equation (Equation 37)	−1.29	−1.66	−1.78	−2.24	−2.44

The order reduction methods below the solid line are designed by our framework.

**Table 5 entropy-25-00535-t005:** Comparison of energy, PSNR and calculation time achieved by different order reduction methods, averaged over the benchmark [1,2]. We highlight the best performance in bold.

Methods	Energy ($\times 10^{4}$)			PSNR			Time ($\times 10^{2}$ s)		
	σ=15	σ=20	σ=25	σ=15	σ=20	σ=25	σ=15	σ=20	σ=25
HOCR Equation (Equation 7)	**5.24**	**4.74**	**4.45**	27.02	25.18	24.05	3.51	3.76	3.62
LogR-1 Equation (Equation 8)	7.36	7.77	7.02	25.52	21.91	22.17	5.47	5.69	5.41
LogR-2 Equation (Equation 9)	5.27	4.83	4.16	27.09	25.36	24.30	3.52	3.74	3.63
LinR Equation (Equation 10)	**5.24**	**4.74**	**4.45**	27.02	25.18	24.05	3.51	3.78	3.62
SqrtR Equation (Equation 11)	8.08	8.31	8.62	25.20	21.68	21.35	9.08	9.02	8.66
GR-1 Equation (Equation 14)	8.13	9.26	10.55	26.18	24.01	22.24	4.09	4.35	4.19
GR-2 Equation (Equation 15)	8.13	9.26	10.55	26.18	24.01	22.24	4.08	4.35	4.18
GR-3 Equation (Equation 16)	5.28	4.83	4.59	27.08	25.31	24.28	5.59	5.63	5.61
SR-1 Equation (Equation 23)	6.28	6.16	5.82	26.05	23.53	23.04	5.92	5.76	5.84
SR-2 Equation (Equation 24)	6.03	6.07	6.38	27.10	25.45	24.16	3.89	3.82	3.90
SR-3 Equation (Equation 25)	6.57	6.39	6.46	26.20	24.38	23.42	5.57	5.41	5.48
SR-4 Equation (Equation 26)	5.27	4.83	4.58	27.09	25.36	24.30	3.47	3.48	3.56
SR-5 Equation (Equation 27)	5.27	4.83	4.58	27.09	25.36	24.30	3.47	3.47	3.55
SR-6 Equation (Equation 28)	5.27	4.83	4.58	27.09	25.36	24.30	3.47	3.47	3.55
MR-1 Equation (Equation 31)	6.31	6.24	6.30	25.94	23.92	22.88	6.63	6.65	6.77
MR-2 Equation (Equation 32)	6.51	6.50	6.47	25.70	23.44	22.53	5.27	5.36	5.39
MR-3 Equation (Equation 33)	5.27	4.82	4.58	27.09	25.35	24.30	**3.46**	**3.45**	**3.53**
MR-4 Equation (Equation 36)	6.29	6.11	6.04	26.17	24.14	23.21	7.59	7.69	7.61
MR-5 Equation (Equation 37)	5.58	5.29	5.25	**27.17**	**25.58**	**24.48**	7.60	7.72	7.62

The order reduction methods below the solid line are designed by our framework.

## Data Availability

The data presented in this study are available on request from the corresponding author.

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
