# Peer review of "An Order Reduction Design Framework for Higher-Order Binary Markov Random Fields"

_entropy, 2023, doi:10.3390/e25030535_

Round 1

Reviewer 1 Report

The paper deals with Image analysis based. In the present form the paper needs improvements. Starting from the introduction authors should provide to the reader a wider picture on the reserach domain in order to better contestualise the propsoed approach. Some more references should be added.

Regarding the methodological section the theoretical framework is well presented even if the integration of a graphical flowchart could support a more direct readibility of the overall methodology. For what concerns the eXperimental validation some more details regarding the experimental settings could better support the reliability of the performance anaslysis 

Reviewer 3 Report

The manuscript entitled " An Order Reduction Design Framework for Higher-Order Binary Markov Random Fields " has been investigated in detail. The topic addressed in the manuscript is potentially interesting and the manuscript contains some practical meanings, however, there are some issues that should be addressed by the authors:

1-    The Introduction section needs a major revision in terms of providing a more accurate and informative literature review and the pros and cons of the available approaches and how the proposed method is different comparatively. Also, the motivation and contribution should be stated more clearly.

2-    The importance of the design carried out in this manuscript can be explained better than in other important studies published in this field. I recommend the authors review other recently developed works

3-    The authors should clearly emphasize the contribution of the study. Please note that the up-to-date of references will contribute to the up-to-date of your manuscript.

https://doi.org/10.3390/math10173172;

https://doi.org/10.1016/j.aej.2020.05.006

4-What makes the proposed method suitable for this unique task? What new development to the proposed method have the authors added (compared to the existing approaches)? These points should be clarified.

5- The performance of the proposed method should be better analyzed, commented and visualized in the numerical section, Stability and convergence of the method should display. Figure (3) surface plot right hand why to error tack place exactly in the corner?

Round 2

Reviewer 2 Report

I believe the authors have answered my questions satisfactorily. 

Reviewer 3 Report

All requrmment have bben done